# Influence of Depression and Anxiety on Hemodialysis Patients: The Value of Multidisciplinary Care

**DOI:** 10.3390/ijerph18073544

**Published:** 2021-03-29

**Authors:** Carlos J. Delgado-Domínguez, Sergio Sanz-Gómez, Ana López-Herradón, Beatriz Díaz Espejo, Olaya Lamas González, Macarena de los Santos Roig, Isabel Berdud Godoy, Abraham Rincón Bello, Rosa Ramos Sánchez

**Affiliations:** 1Centro de Hemodiálisis San Rafael, Fresenius Medical Care Services Andalucía (FMCSA), Calle Julio Arteche, 1, 14005 Córdoba, Spain; beatriz.diaz@fmc-ag.com (B.D.E.); isabel.berdud@fmc-ag.com (I.B.G.); 2Departamento de Psiquiatría, Facultad de Medicina, Universidad de Sevilla, Avenida Sánchez Pizjuán, s/n, 41009 Sevilla, Spain; ssanz1@us.es; 3Departamento Médico, Fresenius Medical Care España, Ronda de Poniente, 8, 28760 Madrid, Spain; ana.lopez@fmc-ag.com (A.L.-H.); olaya.lamas@gmail.com (O.L.G.); abraham.rincon@fmc-ag.com (A.R.B.); 30965rrs@gmail.com (R.R.S.); 4Departamento de Metodología de las Ciencias del Comportamiento, Facultad de Psicología, Campus de Cartuja, s/n, Universidad de Granada, 18011 Granada, Spain; dlsantos@ugr.es

**Keywords:** anxiety, depression, psychological inflexibility, hemodialysis, quality of life

## Abstract

Affective disorders promote poorer outcomes in hemodialysis patients. According to the presence or not of depression/anxiety in these patients, aims were to analyze differences in sociodemographic, clinical and/or psychological factors and to identify predictors. One hundred eighty-six hemodialysis patients were classified based on their depression/anxiety status. Basal characteristics showed differences between groups where mainly male sex (Depression: OR 0.2; Anxiety: OR 0.3) albumin (Depression: OR 0.1; Anxiety: OR 0.2) and calcium levels (Depression: OR 0.5; Anxiety: OR 0.4), impaired quality of life (Depression: OR 1.4; Anxiety: OR 1.2) and psychological inflexibility (Depression: OR 1.3; Anxiety: OR 1.2) were associated (all *p* < 0.01) to these mental conditions. Multivariate models showed that worse quality of life (OR 1.3; *p* < 0.001) predicted depression while marital status (with a partner; OR 0.3; *p* = 0.025) and albumin levels (OR 0.1; *p* = 0.027) were protective factors. Depression represented a risk factor for anxiety (OR 1.2; *p* = 0.001), although calcium levels (OR 0.5; *p* = 0.039) would protect this state. Interestingly, psychological inflexibility predicted both disorders (Depression: OR 1.2, *p* < 0.001 and Anxiety: OR 1.1; *p* = 0.002). Results highlight the relevance of well-trained multidisciplinary hemodialysis units to control the influence of these factors on the presence of depression/anxiety, and thus, their impact on the patients’ outcomes.

## 1. Introduction

Affective disorders are well-known risk factors for health and quality of life of patients on hemodialysis (HD) treatment [1]. In the last 20 years, great efforts have been made in the field of nephrology and health psychology to detect and account for the high prevalence of affective disorders in patients with chronic kidney disease (CKD) who should be cared for in HD units. Some works published in the 2000s have already highlighted that depressive patients undergoing HD had a higher mortality rate than those who do not [2,3]. More recently, depressed mood has also been associated with up to four times more of withdrawal from HD treatment and an increased risk of hospitalizations [3,4,5]. Alcohol dependence, abuse of other addictive substances or other mental health problems with the subsequent increased risk of death by suicide are other relevant study variables to be considered in these particular patients [6].

Anxiety, although less studied than depression in the context of CKD, is a psychological disorder that strongly impacts the health state and quality of life of patients on HD treatment [1,4,7]. Therefore, it is considered that this condition is directly related to poorer clinical outcomes derived from HD, promoting less patients’ adherence to their treatment and impairing also the nutritional habits which increases mortality rates [7]. Despite all the evidence highlighting the impact of depression and anxiety in CKD patients, the introduction of health psychologists in HD units staffing is still limited as well as the therapeutic tools to work with these particular patients. This impact needs to be seriously supervised in HD units and even more when it is well documented that many patients, despite years of HD therapy, often continue experiencing anxiety during the weekly sessions they receive [4].

Psychological inflexibility (PI) has been proposed as a process that underlies both disorders and facilitates their emergence and course [8]. This construct is defined as “*a rigid pattern of behavior in which a person directs their actions on the basis of momentary private experiences rather than freely chosen values*” [9]. PI and experiential avoidance are theorized to contribute to the development, maintenance and exacerbation of a wide range of psychological problems. Although the form of problem behaviors differs between disorders, many of them can be conceptualized as sharing common psychological functions [10]. However, in the field of HD, there is no research addressing this key concept in clinical psychology.

Patients with CKD and particularly those undergoing HD, often require mental health support due to anxious-depressive conditions. However, it is uncommon that the presence of multidisciplinary teams in HD units are able to analyze and follow-up on parameters involved in mental health and the quality of life of the patients. Therefore, based on the presence or not of depression or anxiety in our patients, the aims of this study were to compare different sociodemographic, psychological, clinical and HD-related parameters between groups and also to identify predictors for these conditions.

## 2. Materials and Methods

### 2.1. Participants

A multi-centre prospective cross-sectional study was carried out in outpatients in the HD programme at Fresenius Medical Care clinics in the province of Córdoba (Spain). The inclusion criteria were: (1) Ability to understand and speak the Spanish language; (2) Age between 18 and 90 years old; (3) Signed informed consent for the study; (4) HD vintage of at least six months; and (5) Absence of cognitive impairment or severe mental disorder. A total of 130 patients were excluded from the potential sample of the four clinics (N = 316), which resulted in a total sample of 186 subjects (58% of participation rate). All patients had previously been informed about data privacy and had provided written informed consent for the use of their data to conduct scientific research. The Research Ethics Committee from Hospital Universitario Reina Sofía (Córdoba, Spain) authorized this study in June 2017.

### 2.2. Sociodemographic and Clinical Variables

Data on sociodemographic characteristics, Charlson Comorbidity Index (CCI), dialysis modality and quality parameters, concomitant medication and laboratory results were collected from the EuCliD database, belonging to Fresenius Medical Care [11]. All blood samples were drawn pre-dialysis. Kt/V was used as indicator of dialysis quality and monitored in every dialysis treatment using the OCM^®^ (On-line Clearance Monitor; Fresenius Medical Care, Bad Homburg, Germany). The mean of all Kt/V values during the month was obtained. The Body Composition Monitor (BCM^®^; Fresenius Medical Care, Bad Homburg, Germany) estimated body composition parameters, including lean and fat tissue indexes (LTI and FTI) and overhydration, calculated as predialysis weight minus normohydrated weight and adjusted per extracellular water. All BCM^®^ measurements were performed before the dialysis session and after 10 min in decubitus.

### 2.3. Psychological and Quality of Life Variables

Different questionnaires for the assessment of patients’ quality of life and mental state were completed during the months of June and July 2017. During the HD sessions, all these questionnaires were provided and explained to the patients by three staff members previously trained, with a completion time of 20–30 min.

The following instruments were administered:

Coop-Wonca Quality of Life Charts [12]. It is a general questionnaire for the evaluation of health-related quality of life (HRQoL) in 9 dimensions: 1. Physical fitness, 2. Feelings, 3. Daily life activities, 4. Social activities, 5. Changes in health status, 6. Health status, 7. Pain, 8. Social support and 9. Overall quality of life. All these dimensions were referred to the last two weeks and answered based on a Likert scale from 1 to 5, where higher scores reflect poorer perceived health. An overall score was obtained from the sum of all the dimensions excepting item 5 that, due to its bipolar structure, requires a different reading from the rest of the dimensions [13]. The HRQoL instrument showed a suitable Cronbach’s Alpha (0.77).

Acceptance and Action Questionnaire II for HD patients (AAQHD-II). This questionnaire was adapted from the original AAQ-II [14] to the context of HD treatment. It assessed the level of PI in the presence of certain thoughts (items 2 and 3), fears (item 6), treatment guidelines (items 4 and 5), situations related to the treatment of HD (item 1) and the effect of treatment guidelines on the patient’s life (item 7). Scores were obtained by a 5-point Likert type scale (1 = Never true, 2 = Rarely true, 3 = Sometimes true, 4 = Often true, 5 = Always true). Cronbach’s Alpha (0.72) was calculated grouping items into a single factor as in the original questionnaire.

Hospital Anxiety and Depression Scale (HADS), in the Spanish version of Terol et al. [15]. This scale has previously shown adequate psychometric properties in several groups of patients, including those in HD treatment in both international and Spanish studies [16,17]. Therefore, we used this instrument to assess emotional responses to anxiety and depression with or without medical and psychiatric conditions but excluding patients’ somatic-type symptoms [17,18]. Patients’ scores were calculated after the completion of fourteen items classified into two subscales, one for anxiety and one for depression, following a 4-point Likert response format, where the higher the score, the greater the presence of anxiety or depression. The Cronbach’s Alpha (0.83) resulted suitable for both subscales.

### 2.4. Statistical Analysis

The sample was divided after the selection of a cut-off point from the total score of each HADS subscale. This cut-off was set at 8 points for both scales, where scores ≥8 determined patients with presence or a likely presence of depressive or anxious condition.

Results were expressed as mean ± standard deviation (SD) for continuous variables with normal distribution, median [25th percentile–75th percentile] for those with non-normal distribution and % for categorical variables. Sociodemographic, clinical, psychological and quality of life-related variables were compared between groups for both depression and anxiety, using Student’s *t*, Mann–Whitney’s U and Chi-square tests according to the nature of the variables. For the identification of predictive factors for anxiety and depression development, variables presenting statistical differences between groups or clinically relevant were included in univariate and multivariate logistic analyses. Linearity of the logit was tested for each variable included [19].

All statistical analyses were performed using the SPSS statistical package version 19.0 (IBM, Armonk, NY, USA). The level of statistical significance was set at *p* < 0.05.

## 3. Results

### 3.1. Baseline Characteristics of Study Participants

Almost 40% of the 186 patients included in the study were women while the mean age was 70 ± 13.8. Regarding sociodemographic factors, 64.5% of the population was married or sharing their lives with a partner and had a low level of education (51.1% with no studies and 31.2% with primary education). Most of the patients (97.8%) were retired.

According to the groups determined by the HADS subscales scores, 52 patients (27.9%) had or probably had a depressive or an anxiety disorder (the proportions found for both conditions were identical). We also observed that 34% of the patients were taking medication for affective disorders, mainly anxiolytics and/or antidepressants.

### 3.2. Baseline Characteristics of Study Participants by the Presence of Depression

Those cases with identified or probable depression were represented by a higher percentage of women, less proportion of married patients and more comorbid conditions compared to the non-depression group. Regarding the results from the psychological assessments in these depressive patients, higher scores for all HRQoL dimensions, PI and greater anxiety and intake of psychotropic drugs were noticed. Additionally, significant changes were detected in some clinical parameters as decreased values for albumin, sodium and calcium serum levels but also in body composition indexes (FTI increased, LTI decreased) (Table 1).

### 3.3. Baseline Characteristics of Study Participants by the Presence of Anxiety

Within the group of patients with identified or probable anxiety, almost 60% were women, obtained worse HRQoL scores (emphasizing the items Feelings, Activities of daily living, Social activities, Changes in health status and Health status areas) and registered higher PI and presence of depression. This group of patients showed less frequency of arteriovenous fistula as vascular access and slight decreases in hemoglobin, albumin and calcium serum levels compared to those patients without anxiety (Table 2).

### 3.4. Predictors of Depression and Anxiety in HD Patients

Univariate logistic analyses showed the influence of different variables on the depressive (Dep) and anxious states (Anx) in our population. Male sex (Dep: OR 0.2; Anx: OR 0.3), albumin (Dep: OR 0.1; Anx: OR 0.2) and calcium serum levels (Dep: OR 0.5; Anx: OR 0.4) were significantly associated to the presence of these mental disorders (all variables described presented *p*-values < 0.01). Moreover, higher HRQoL scores (Dep: OR 1.4; Anx: OR 1.2) and/or PI (Dep: OR 1.3; Anx: OR 1.2) seemed to prompt both conditions (Table 3 and Table 4).

After the inclusion of all significant variables from the aforementioned analyses and those clinically relevant into multivariate analyses, we confirmed that higher HRQoL scores (OR 1.3; *p* < 0.001) predicted the development of depression in HD patients while having a partner (OR 0.3; *p* = 0.025) and serum albumin levels (OR 0.1; *p* = 0.027) were associated with a lower presence of this condition (Table 3). Depression as comorbidity represented a risk factor for anxiety state (OR 1.2; *p* = 0.001) while calcium serum levels (OR 0.5; *p* = 0.039) seemed to protect this mental condition. Interestingly, we found that the PI was associated to an increased risk of suffering both, depression or anxiety (Dep: OR 1.2, *p* < 0.001 and Anx: OR 1.1; *p* = 0.002) (Table 4).

## 4. Discussion

The present study collected important data regarding prevalence of depression and anxiety in patients with CKD on HD treatment. The results obtained from our study population showed that more than a quarter of the patients evaluated presented with depression and to the same extent anxiety. These rates were similar to those found in other previous studies [2,20,21,22,23] and confirmed the actual need to take an in-depth look into the psycho-affective aspects of HD patients [24]. In our analysis, women showed a higher rate of depression than men, a common finding as women are more likely to suffer from affective disorders [25]. However, these gender differences disappeared when we included in the multiple regression analyses greater determinants as marital state, quality of life and PI. 

Serum albumin levels were determined as a risk factor of depression in our multiple regression analysis, a finding which could support two previous studies from Taiwan [26] and Brazil that confirmed the clinical relevance of [27] the influence of depression on appetite as a rational for a significantly worse nutritional status in HD patients. In this sense, our results would also reinforce the suggestion of Friend et al. to take seriously the association between depression and nutritional status in HD units due to its additive effect on patients’ mortality [28]. Furthermore, as proposed by other authors like Abdulan et al., depression would be a consequence of malnutrition especially in elderly patients undergoing HD [29].

Among the clinical parameters associated to altered mental status, we observed decreased serum calcium and sodium levels. It seems that decreased sodium levels could elicit depressive symptoms in animal models [30]. In addition, some evidences described depression and stress as possible modulators acting on sodium levels through different metabolic pathways [31]. As for calcium, previous research on patients with psychiatric disorders did not find solid relations between serum calcium levels and the presence of depression [32,33]. However, a recent study has pointed to a possible association of low-calcium dietary intake and the presence of both depression and anxiety in young adults [34]. In our study, biochemical changes related to these electrolytes were showed as a risk factor in the univariate analyses but not in the multivariate models when they were adjusted by other variables. We suggest that fluctuations in sodium and calcium levels are common and were more likely related to the patients’ nutritional status. Even so, it is clear that further studies are needed to elucidate the influence of balanced/unbalanced electrolytes serum levels on the mental health in patients with CKD.

Detection of a depressive disorder in patients with CKD can be challenging since symptoms may be mistaken or a consequence from uremia [21]. Thus, an effective and early approach to these conditions’ results are essential as untreated depression: (1) lead to poorer outcomes for these patients and (2) could also affect both, the adherence to the different treatments for CKD and their commitment to implement and maintain the recommendations for healthy living habits [35,36]. It would be of interest for future studies to quantify how much lifestyle factors (physical activity, different nutritional habits… etc.) and the quality of dialysis could influence changes in body mass composition in HD patients with or without altered mental status [37]. 

Unlike depression, the incidence and role of anxiety in CKD has been less studied [7]. However, in our study, this condition was associated with a decline in quality of life standards. Previous studies already highlighted this concern in patients with CKD [1,7], finding that anxiety affected quality of life negatively and directly, and not only as a by-product of a depressive picture. However, the multivariate regression of the present study shows how quality of life loses its predictive power when analyzed including other factors, such as PI. 

Regarding the use of medication for affective disorders, there were no differences between groups which could be interpreted as these drugs did not affected the patients’ emotional state. However, it should be noteworthy that patients can show high scores on anxiety and depression scales even when they are following a concomitant treatment with anxiolytics and/or antidepressants. Pharmacological treatments may be effective for the management of the most physiological symptoms derived from affective disorders, but cannot result enough to tackle their cognitive and emotional components [38]. This condition could be much more aggravated in HD patients who feel already anxious by their own HD treatment and other factors like transportation needs, or experience adverse intra- or post-dialysis events.

Different techniques (cognitive-behavioral interventions, mindfulness-based therapies or psychosocial interventions) have shown their effectiveness in improving the psycho-affective state of patients with CKD, the adherence to their medical prescriptions and the HD clinical outcomes [39,40,41,42,43]. In this line, our results reveal the considerable number of variables (clinical, psychological, social, emotional) that surround the patient on HD treatment. The accumulated evidence until now on the relevance and value of psychological care in patients with CKD is driving, although more slowly than desired, the implementation of Psychonephrology as a bridging discipline. The influence of mediators such as the quality of life, the PI or other biochemical parameters in two common affective disorders like depression or anxiety force to the development of new therapeutic approaches on this field. Based on our results, we truly believe that the integration of psychotherapies as part of the current HD treatment programs would benefit patients’ clinical and psychological parameters, and thus, their clinical outcomes.

This is the largest Spanish study on the subject so far and analyzes 186 HD patients in four clinics. Multidisciplinary care in clinical practice has allowed a systematic collection of clinical, nutritional and psychological parameters with a high degree of standardization and good data quality. In addition, it has focused on transdiagnostic aspects of clinical psychology, such as PI. Although PI has not received attention in this type of population, it is especially relevant due to its idiosyncratic problems such as lack of adherence to treatment. However, the study also has some limitations. Firstly, due to its cross-sectional design, we could analyze associations between variables but could not set a causal link. Secondly, a higher sample size and the inclusion of patients from other regions of Spain would confirm and provide more precise data. Lastly, this study was not designed to assess the influence of other possible confounding factors such as chronic pain, specific comorbidities not related to CKD, inflammation markers and/or other mental disorders. These factors should be included in future studies.


In conclusion, our results evidence the actual need of HD units formed by multidisciplinary teams trained not only in CKD management but also in the assessment of mental health status as part of an integrated HD treatment. Moreover, this is the first time that the PI is associated directly with the risk of development of both mental disorders, depression or anxiety, in HD patients.

## Figures and Tables

**Table 1 ijerph-18-03544-t001:** Baseline characteristics of study population according to the depressive state.

		Non-Depressive Group	Depressive Group	*p*-Value
		N = 134	N = 52	
Sociodemographic				
Hemodialysis clinic	FMC-San Rafael	47.0%	36.5%	0.506
	FMC-Pintor Gª Güijo	19.4%	26.9%	
	FMC-Cabra	26.9%	26.9%	
	FMC-Palma del Río	6.7%	9.6%	
Sex, women		29.9%	63.5%	<0.001
Age (years)		69.0 ± 14.6	73.1 ± 11.1	0.070
Marital status	Married/with domestic partner	70.1%	50.1%	0.010
Educational level	No schooling	49.3%	55.8%	0.398
	Primary school	34.3%	23.1%	
	High school	9.7%	15.4%	
	University	6.7%	5.8%	
Employment status	Unknown	0.7%	0.0%	0.452
	Unemployed	2.2%	0.0%	
	Retired	97.0%	100.0%	
Quality of life				
Health-related Quality of Life (HRQoL)	Physical Fitness	3 (3–4)	4 (4–5)	<0.001
	Feelings	2 (1–3)	3 (2–4)	<0.001
	Daily Life Activities	2 (1–3)	3 (3–4)	<0.001
	Social Activities	1 (1–2)	3 (1,2–4)	<0.001
	Changes in Health Status	3 (2–3)	3 (3–4)	0.002
	Health Status	3 (3–4)	4 (3–4)	<0.001
	Pain	2 (1–4)	4 (2.2–4)	<0.001
	Social Support	2 (2–3)	3 (2–3.7)	0.001
	Quality of Life	2 (2–3)	3 (3–4)	<0.001
	Total score	19 (16–22)	25.5 (23–30)	<0.001
Psychological				
Psychological Inflexibility (AAQHD-II)	Total score	13 (10–16)	19 (16–22)	<0.001
HADS subscale, Anxiety	Total score	4 (2–6)	8 (6–12)	<0.001
Clinical				
Psychotropic drugs use		26.9%	53.8%	0.001
	Nº psychotropic drugs	1 (1–2)	1 (1–2)	0.645
	Anxiolytics	100.0%	94.4%	0.274
	Benzodiazepines	100.0%	100.0%	-
	Antidepressants	93.3%	100.0%	0.309
Dialysis vintage (months)		55.5 (22.7–112.5)	49.5 (34–100)	0.960
Renal transplant waiting list, yes		11.2%	9.6%	0.755
Vascular access, arteriovenous fistula		82.8%	71.2%	0.076
Charlson Comorbidity Index (CCI), score		4 (2–5)	4 (3–6)	0.003
Kt/V		1.8 ± 0.4	2 ± 0.3	0.268
K (mEq/L)		5.2 ± 0.8	5.1 ± 0.8	0.244
Na (mEq/L)		139.3 ± 2.7	138.4 ± 2.6	0.033
Hemoglobin (g/dL)		11.4 ± 1.2	11.1 ± 1.5	0.190
Transferrin saturation (%)		22 (17–29)	18.5 (16- 27.7)	0.128
C-reactive protein (mg/l)		6.7 (2.1–15.6)	8.5 (4.4–16.8)	0.102
Pre-HD relative overhydration (%)		11.3 (5.5–15.2)	11.6 (6.1- 19.5)	0.294
Pre-HD systolic blood pressure (mmHg)		133.6 ± 21.9	132.1 ± 27.9	0.701
Albumin (g/dL)		3.7 ± 0.3	3.5 ± 0.3	<0.001
Lean Tissue Index (LTI) (Kg/m^2^)		11.8 ± 3.01	10.5 ± 2.7	0.002
Fat Tissue Index (FTI) (Kg/m^2^)		15.3 ± 5.7	18.6 ± 8.0	0.017
Ca (mg/dL)		8.9 ± 0.6	8.7 ± 0.6	0.017
*p* (mg/dL)		4 ± 1.2	4.2 ± 1.5	0.350

Note: Values expressed as mean ± SD, median (P25–P75) or frequency (%). Depressive group was defined by patients presenting scores ≥8 points in HADS depression subscale. Similarly, non-depressive group corresponds to those patients who scored <8 points in HADS.

**Table 2 ijerph-18-03544-t002:** Baseline characteristics of study population according to the anxious state.

		Non-Anxious Group	Anxious Group	*p*
		N = 134	N = 52	
Sociodemographics				
Hemodialysis clinic	FMC-San Rafael	45.5%	40.4%	0.563
	FMC-Pintor Gª Güijo	22.4%	19.2%	
	FMC-Cabra	26.1%	28.8%	
	FMC-Palma del Río	6%	11.5%	
Sex, women		31.3%	59.6%	<0.001
Age (years)		70.4 ± 13.2	69.6 ± 15.5	0.754
Marital status	Married/with domestic partner	66.4%	59.6%	0.384
Educational level	No schooling	48.5%	57.7%	0.477
	Primary school	34.3%	23.1%	
	High school	10.4%	13.5%	
	University	6.7%	5.8%	
Employment status	Unknown	0.7%	0%	0.806
	Unemployed	1.5%	1.9%	
	Retired	97.8%	98.1%	
Quality of life				
Health-related Quality of Life (HRQoL)	Physical Fitness	4 (3–4)	4 (3–4)	0.072
	Feelings	1 (1–2)	3,5 (3–4)	<0.001
	Daily Life Activities	2 (1–3)	3 (2.25–4)	<0.001
	Social Activities	1 (1–2)	2 (1–4)	<0.001
	Changes in Health Status	3 (2.7–3)	3 (3–4)	0.011
	Health Status	3 (3–4)	4 (3–4)	0.051
	Pain	3 (1–4)	3 (1–4)	0.162
	Social Support	2 (2–3)	2.5 (2–3,7)	0.237
	Quality of life	2 (2–3)	3 (2–4)	<0.001
	Total score	20 (16.7–23)	24.5 (21.2–28.7)	<0.001
Psychological				
Psychological Inflexibility (AAQHD-II)	Total score	13 (10–17)	18 (15–22)	<0.001
HADS subscale, Depression	Total score	4 (2–6)	8 (5.2–10)	<0.001
Clinical				
Psychotropic drugs use		30.6%	44.2%	0.079
	Nº psychotropic drugs	1 (1–2)	1 (1–2)	0.972
	Anxiolytics	96.2%	100%	0.474
	Benzodiazepines	100%	100%	-
	Antidepressants	100%	90.9%	0.181
Dialysis vintage (months)		56.5 (26–104)	45 (28.25–104.5)	0.467
Renal transplant waiting list, yes		10.4%	11.5%	0.829
Vascular Access, arteriovenous fistula		85.1%	65.4%	0.003
Charlson Comorbility Index (CCI) score		4 (2–5)	4 (3–5)	0.387
Kt/V		1.9 ± 0.4	1.9 ± 0.4	0.571
K (mEq/L)		5.2 ± 0.8	5.2 ± 0.8	0.626
Na (mEq/L)		139.2 ± 2.9	138.8 ± 2.3	0.387
Hemoglobin (g/dL)		11.5 ± 1.4	11.0 ± 1.2	0.024
Transferrin saturation (%)		22 (16–29.2)	21 (16.2–28)	0.758
C-reactive protein (mg/L)		6.8 (2.2–15.1)	9.3 (3.7–19.8)	0.073
Pre-HD relative overhydration (%)		11.5 (5.3–15.3)	11.1 (6.3–18.9)	0.458
Pre-HD systolic blood pressure (mmHg)		134.5 ± 21.2	129.9 ± 29.0	0.242
Albumin (g/dL)		3.7 ± 0.3	3.6 ± 0.3	0.003
Lean tissue index (LTI) (Kg/m^2^)		11.5 ± 3.0	11.2 ± 2.9	0.472
Fat tissue index (FTI) (Kg/m^2^)		15.6 ± 5.9	17.8 ± 7.9	0.129
Ca (mg/dL)		8.9 ± 0.6	8.6 ± 0.7	0.003
*p* (mg/dL)		4 ± 1.2	4.2 ± 1.4	0.262

Note: Values expressed as mean ± SD, median (P25-P75) or frequency (%). Anxious group was defined by patients who obtained scores ≥8 points in HADS anxiety subscale. Similarly, non-anxious group is represented by patients whose scores were <8 points in the scale.

**Table 3 ijerph-18-03544-t003:** Univariate and multivariate logistic analyses. Regression models performed to identify predictive factors for depressive state in patients on HD treatment.

	Univariate Logistic Regression		Multivariate Logistic Regression	
	OR (CI, 95%)	*p*-Value	OR (CI, 95%)	*p*-Value
Sex, men	0.2 (0.1–0.5)	<0.001	1.1 (0.3–3.7)	0.869
Marital status, married/with domestic partner	0.4 (0.2–0.8)	0.011	0.3 (0.1–0.9)	0.025
Health-related Quality of Life (HRQoL score)	1.4 (1.3–1.6)	<0.001	1.3 (1.2–1.5)	<0.001
Psychological inflexibility (AAQHD-II score)	1.3 (1.2–1.4)	<0.001	1.2 (1.1–1.4)	<0.001
HADS subscale score, anxiety	1.4 (1.2–1.5)	<0.001	1.1 (0.9–1.3)	0.167
Charlson Comorbility Index (CCI) score	1.3 (1.1–1.6)	0.004	1.4 (1–1.9)	0.051
Psychotropic drugs users	3.2 (1.6–6.2)	0.001	0.5 (0.2–1.8)	0.318
Albumin (g/dL)	0.1 (0–0.4)	<0.001	0.1 (0.2–0.8)	0.027
Na (mEq/L)	0.9 (0.8–1)	0.035	1.0 (0.8–1.2)	0.681
Ca (mg/dL)	0.5 (0.3–0.9)	0.020	0.7 (0.3–2.3)	0.721
Lean tissue index (LTI) (Kg/m^2^)	0.8 (0.7–1)	0.009	0.9 (0.7–1)	0.097
Fat tissue index (FTI) (Kg/m^2^)	1.1 (1–1.1)	0.004	1 (0.9–1)	0.659

Note: OR, Odds ratio; CI, Confidence interval.

**Table 4 ijerph-18-03544-t004:** Univariate and multivariate logistic analyses. Regression models performed to identify predictive factors for anxiety state in patients on HD treatment.

	Univariate Logistic Regression		Multivariate Logistic Regression	
	OR (CI, 95%)	*p*-Value	OR (CI, 95%)	*p*-Value
Sex, men	0.3 (0.2–0.6)	0.001	0.6 (0.3–1.41)	0.267
Age (years)	1 (0.9–1)	0.752	1 (0.9–1)	0.259
Health-related Quality of Life (HRQoL)	1.2 (1.1–1.3)	<0.001	1.1 (0.9–1.2)	0.286
Psychological Inflexibility (CAAH-II)	1.2 (1.1–1.3)	<0.001	1.1 (1–1.2)	0.002
HADS subscale, depression	1.3 (1.2–1.5)	<0.001	1.2 (1.1–1.4)	0.001
Psychotropic drugs users	1.8 (0.9–3.5)	0.081	1 (0.4–2.5)	0.989
HD vintage (months)	1 (0.9–1)	0.584	1 (0.9–1)	0.778
Waiting list for kidney transplant, included	0.9 (0.3–2.5)	0.829	1.1 (0.2–4.9)	0.895
Vascular Access, catheter	3 (1.4–6.3)	0.004	1.6 (0.5–5.2)	0.406
Kt/V	1.3 (0.6–2.7)	0.569	1 (0.9–1)	0.747
Hemoglobin (g/dL)	0.8 (0.6–1)	0.026	0.7 (0.5–1)	0.053
C-reactive protein (mg/l)	1 (0.9–1)	0.094	1 (0.9–1)	0.275
Albumin (g/dL)	0.2 (0.1–0.6)	0.005	0.7 (0.1–3)	0.626
Ca (mg/dL)	0.4 (0.3–0.8)	0.004	0.5 (0.3–1)	0.039
Pre-HD systolic blood pressure (mmHg)	1 (0.9–1)	0.418	1 (0.9–1)	0.403

Note: OR, Odds ratio; CI, Confidence interval.

## Data Availability

Data are available upon reasonable request to the corresponding author.

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
