# Peer review of "Influence of Depression and Anxiety on Hemodialysis Patients: The Value of Multidisciplinary Care"

_ijerph, 2021, doi:10.3390/ijerph18073544_

Round 1

Reviewer 1 Report

This is an interesting study and I don’t have any major comment other than it is difficult to understand the manuscript in the current form. The abstract and the manuscript should be modified so that the relevance of the study is clearer. This study is much more interesting than the Abstract and text show.  

Author Response

Manuscript ID: ijerph-1114677

Title: Influence of depression and anxiety on hemodialysis patients: the value of multidisciplinary care.

Round #2

ANSWERS TO MINOR REVISION

Date: 23th Marc 2021

Reviewer #1

This is an interesting study and I don’t have any major comment other than it is difficult to understand the manuscript in the current form. The abstract and the manuscript should be modified so that the relevance of the study is clearer. This study is much more interesting than the Abstract and text show.

Thank you so much for your comments and appreciations. We have updated the Abstract however, the limitation of just 200 words makes quite complicated to resume fairly the content of the manuscript. We have tried to synthetize the body and emphasized the final conclusion. In addition, some changes have been included in the text in order to improve redaction and clarity of the manuscript.

Reviewer 2 Report

The article "Influence of depression and anxiety on hemodialysis patients: the value of multidisciplinary care" by Carlos Jesús Delgado-Domínguez et al. evaluates the emotional state in 186 Chronic Kidney Disease (CKD) patients on hemodialysis and identifies sociodemographic, psychological, clinical, and HD-related predictors of anxiety and depression. This is an interesting study that underscores the value of multidisciplinary care aimed at reducing biochemical and physiological abnormalities, as well as psychological disorders experienced by CKD patients. Here are some comments to improve the manuscript.

The authors mention that 34% of CKD patients were taking medication for affective disorders, anxiolytics, and/or antidepressants (Line 155). However, it is surprising that these medications seem to have no impact on the patients' emotional state. Could the authors discuss more in depth the role of these medications and the impact of their results on patient care? Additionally, the manuscript would greatly benefit from stating the study's strengths and limitations in the discussion section.

Define abbreviations when first mentioned and use them consistently throughout the manuscript (ex. Chronic kidney disease appears first in Line 38 but is abbreviated as CKD for the first time in Line 47).

Author Response

Reviewer #2

The article "Influence of depression and anxiety on hemodialysis patients: the value of multidisciplinary care" by Carlos Jesús Delgado-Domínguez et al. evaluates the emotional state in 186 Chronic Kidney Disease (CKD) patients on hemodialysis and identifies sociodemographic, psychological, clinical, and HD-related predictors of anxiety and depression. This is an interesting study that underscores the value of multidisciplinary care aimed at reducing biochemical and physiological abnormalities, as well as psychological disorders experienced by CKD patients. Here are some comments to improve the manuscript.

The authors mention that 34% of CKD patients were taking medication for affective disorders, anxiolytics, and/or antidepressants (Line 155). However, it is surprising that these medications seem to have no impact on the patients' emotional state. Could the authors discuss more in depth the role of these medications and the impact of their results on patient care?

Thanks for your suggestion, regarding this topic we have included a new paragraph in Page 8 of 13 (Line 254) as part of the Discussion.

Additionally, the manuscript would greatly benefit from stating the study's strengths and limitations in the discussion section.

Strenghts and limitations have been added in Page 9 of 13 (Line 279).

Define abbreviations when first mentioned and use them consistently throughout the manuscript (ex. Chronic kidney disease appears first in Line 37 but is abbreviated as CKD for the first time in Line 45).

Thanks for noticing, all text has been reviewed and corrected accordingly.

This manuscript is a resubmission of an earlier submission. The following is a list of the peer review reports and author responses from that submission.